# REFLECTION-BASED WORD ATTRIBUTE TRANSFER

## ABSTRACT

We propose a word attribute transfer framework based on *reflection* to obtain a word vector with an inverted target attribute for a given word in a word embedding space. Word embeddings based on Pointwise Mutual Information (PMI) represent such analogic relations as $\overrightarrow{king} - \overrightarrow{man} + \overrightarrow{woman} \approx \overrightarrow{queen}$. These relations can be used for changing a word's attribute from *king* to *queen* by changing its gender. This attribute transfer can be performed by subtracting a difference vector $\overrightarrow{man} - \overrightarrow{woman}$ from $\overrightarrow{king}$ when we have explicit knowledge of the gender of given word *king*. However, this knowledge cannot be developed for various words and attributes in practice. For transferring *queen* into *king* in this analogy-based manner, we need to know that *queen* denotes a female and add the difference vector to it. In this work, we transfer such binary attributes based on an assumption that such transfer mapping will become identity mapping when we apply it twice. We introduce a framework based on reflection mapping that satisfies this property; *queen* should be transferred back to *king* with the same mapping as the transfer from *king* to *queen*. Experimental results show that the proposed method can transfer the word attributes of the given words, and does not change the words that do not have the target attributes.

## 1 INTRODUCTION

Distributed representation (Hinton et al., 1984) is a kind of data representation that can capture data similarities in a vector space. In natural language processing, various studies have been conducted on word embeddings (Mikolov et al., 2013a;b; Pennington et al., 2014; Peters et al., 2018; Bojanowski et al., 2017). Word embedding models, such as skip-gram with negative sampling (SGNS) (Mikolov et al., 2013b) or GloVe (Pennington et al., 2014), capture some analogic relations, such as $\overrightarrow{king} - \overrightarrow{man} + \overrightarrow{woman} \approx \overrightarrow{queen}$. Previous work offer theoretical explanation based on Pointwise Mutual Information (PMI; Church & Hanks (1990)) for maintaining the analogic relations in word vectors (Levy & Goldberg, 2014b; Arora et al., 2016; Gittens et al., 2017; Ethayarajh et al., 2019; Allen & Hospedales, 2019).

These relations can be used for transferring a certain attribute of a word, such as changing *king* into *queen* by transferring the gender. This task, which is called *word attribute transfer*, enables us to rewrite *He is a boy* as *She is a girl*. Word attribute transfer is expected to be applicable for natural language inference and data augmentation in natural language processing. The above analogic relations can be used, including adding difference vector $\overrightarrow{woman} - \overrightarrow{man}$ to $\overrightarrow{king}$ to transfer $\overrightarrow{king}$ to $\overrightarrow{queen}$. This operation requires the explicit knowledge whether an input word is male or female; we have to add a difference vector to a male word and subtract it from a female word for a gender transfer. We also have to avoid changing words without any gender attributes, such as *is* and *a* in the example above. Thus, analogy-based word attribute transfer requires explicit knowledge of word attributes, such as *king* is male, *queen* is female, and *is* has no gender attribute. Developing such knowledge is very difficult for various words and attributes in practice.

In this study, we propose a novel framework based on *reflection*, which enables word attribute transfer by a single reflection-based mapping for a certain attribute. Reflection in geometry is a mapping that exchanges the locations of two vectors in a Euclidean space by a hyperplane called a *mirror*, which satisfies the above desired property: working as identity mapping when it is applied twice and when it is applied to vectors on the mirror. We apply this reflection mapping to the problem of word attribute transfer by estimating an appropriate mirror that maps word pairs with a binary target

attribute (e.g., male and female) and keeps the other words without that attribute using training data. We also extend this approach by introducing *parameterized mirrors*, which work as different mirrors based on the given input words, to overcome a limitation using a single fixed mirror to represent complex transfer mappings for different words. Experimental results show that the reflection-based method enables such transfers, achieves comparable performance to analogy-based methods with explicit attribute knowledge, even though our proposed method does not use such knowledge.

The following are the contributions of this paper:

- We propose a novel representation learning framework that obtains a vector with an inverted attribute in embedding space without explicit attribute knowledge of the given word.
- Our proposed reflection-based word attribute transfer enables us to transfer word attributes in up to 76% for words with target attributes and to avoid changing words without target attributes in over 99% in our experiments.

## 2 WORD ATTRIBUTE TRANSFER

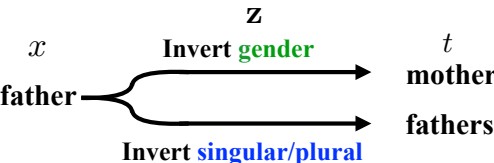

**Figure 1:** Given word vector $\mathbf{v}_x$ and attribute one-hot vector $\mathbf{z}$, word attribute transfer predicts word vector $\mathbf{v}_t$, which is the inverted attribute of $\mathbf{v}_x$.

Let $x$ denote a word and let $\mathbf{v}_x$ denote its vector representation. Here we assume that $\mathbf{v}_x$ is learned in advance with an embedding model such as SGNS. In this task, we have two inputs, word $x$ and one-hot vector $\mathbf{z}$, representing a certain target attribute, and one output, return word $t$ with the inverted attribute of $x$ for $\mathbf{z}$. Let $\mathcal{A}$ denote a set of a triplet $(x, t, \mathbf{z})$, e.g., $(man, woman, \mathbf{z}_{\text{gender}}) \in \mathcal{A}$. Let $\mathcal{N}$ denote a set of words without attribute $\mathbf{z}$, e.g., $apple \in \mathcal{N}$, when $\mathbf{z}$ represents gender. The purpose of this task is to transfer $\mathbf{v}_x$ to $\mathbf{v}_t$ by transfer function $f_{\mathbf{z}}$ that inverts attribute $\mathbf{z}$ of $\mathbf{v}_x$. In other words, output $\mathbf{v}_y$ should be close to the vector of corresponding target word $\mathbf{v}_t$, which is typically the nearest neighbor of $\mathbf{v}_t$ in the word embedding space.

$$\mathbf{v}_t \approx \mathbf{v}_y = f_{\mathbf{z}}(\mathbf{v}_x). \tag{1}$$

Note that mapping $f_{\mathbf{z}}$ transfers word $x$ if it has target attribute $\mathbf{z}$; otherwise $f_{\mathbf{z}}$ works as identity mapping. For instance with $\mathbf{z}_{gender}$, given input word *man*, gender attribute transfer $f_{\mathbf{z}_{gender}}(\mathbf{v}_{man})$ should result in a vector close to $\mathbf{v}_{woman}$. Given input word *apple* as $x$, the results should be $\mathbf{v}_{apple}$.

## 3 ANALOGY-BASED WORD ATTRIBUTE TRANSFER

Analogy is a general idea for realizing attribute transfer. Several PMI-based word embedding methods (Mikolov et al., 2013c; Linzen, 2016) tackled to embed words into word embedding space to capture the analogic relations. An embedded vector with SGNS or GloVe captures analogic relations (Levy & Goldberg, 2014a; Mikolov et al., 2013c; Linzen, 2016). For instance, $\mathbf{v}_{queen}$ is near the vector obtained on the right side of Eq. 2. By rearranging Eq. 2, Eq. 3 is obtained:

$$\mathbf{v}_{queen} \approx \mathbf{v}_{king} - \mathbf{v}_{man} + \mathbf{v}_{woman}, \tag{2}$$

$$\approx \mathbf{v}_{king} - (\mathbf{v}_{man} - \mathbf{v}_{woman}). \tag{3}$$

We can transfer the gender attribute by subtracting difference vector $\mathbf{v}_{man} - \mathbf{v}_{woman}$ from input word vectors, e.g., $\mathbf{v}_{king}$. The analogy-based transfer function is

$$f_{\mathbf{z}}(\mathbf{v}_x) = \begin{cases} \mathbf{v}_x - \mathbf{d} & \text{if} \quad x \in \mathcal{M}, \\ \mathbf{v}_x + \mathbf{d} & \text{if} \quad x \in \mathcal{F}, \end{cases} \tag{4}$$

where $\mathbf{d}$ is a difference vector of the given word pair such as *man* and *woman*, $\mathcal{M}$ is a set of words having a target attribute, and $\mathcal{F}$ is a set of words having an inverse attribute, for example, $man \in \mathcal{M}$ and $woman \in \mathcal{F}$ for gender attributes. Eq. 4 indicates that the operation changes depending on whether input word $x$ belongs to $\mathcal{M}$ or $\mathcal{F}$. For gender words, we subtract difference vector $\mathbf{d}$ if $x$ is male, and add it if $x$ is female. Therefore, we need such explicit knowledge. However, this knowledge cannot be developed for various words and attributes in practice.

## 4 REFLECTION-BASED WORD ATTRIBUTE TRANSFER

### 4.1 IDEALIZED TRANSFER WITHOUT EXPLICIT KNOWLEDGE

What is an idealized transfer function $\phi_{\mathbf{z}}$ for the word attribute transfer? The following are the idealized natures of such a transfer function:

$$\mathbf{v}_m = \phi_{\mathbf{z}}(\mathbf{v}_w), \tag{5}$$

$$\mathbf{v}_w = \phi_{\mathbf{z}}(\mathbf{v}_m), \tag{6}$$

where $m \in \mathcal{M}$ and $w \in \mathcal{F}$. This function $\phi_{\mathbf{z}}$ enables to transfer a word without explicit knowledge. Function $\phi_{\mathbf{z}}$ transfers $\mathbf{v}_m$ to $\mathbf{v}_w$ and $\mathbf{v}_w$ to $\mathbf{v}_m$ without such explicit knowledge as $m \in \mathcal{M}$ and $w \in \mathcal{F}$. By combining Eqs. 5 and 6, we obtain the following formula:

$$\forall m \in \mathcal{M}, \qquad \mathbf{v}_m = \phi_{\mathbf{z}}(\,\phi_{\mathbf{z}}(\mathbf{v}_m)\,), \tag{7}$$

and

$$\forall w \in \mathcal{F}, \qquad \mathbf{v}_w = \phi_{\mathbf{z}}(\,\phi_{\mathbf{z}}(\mathbf{v}_w)\,). \tag{8}$$

Hence, the idealized transfer function is a mapping that becomes an identity mapping when we apply it twice for any $\mathbf{v}$. Such a mapping is called *involution* in geometry. For example, $\phi\colon \mathbf{v} \mapsto -\mathbf{v}$ is one example of an involution. Note that the identity map itself, such as $\phi\colon \mathbf{v} \mapsto \mathbf{v}$, is excluded from the involution.

### 4.2 REFLECTION

A reflection is an involution that reverses the location between two vectors in a Euclidean space through an affine hyperplane (mirror). Reflection is an idealized function because every point returns to its original location when reflection is applied twice:

$$\forall \mathbf{v} \in \mathbb{R}^n, \qquad \mathbf{v} = Ref_{\mathbf{a},\mathbf{c}}(\,Ref_{\mathbf{a},\mathbf{c}}(\mathbf{v})\,). \tag{9}$$

Given vector $\mathbf{v}$ in Euclidean space $\mathbb{R}^n$, the formula for the reflection in the mirror is given:

$$Ref_{\mathbf{a},\mathbf{c}}(\mathbf{v}) = \mathbf{v} - 2\frac{(\mathbf{v} - \mathbf{c}) \cdot \mathbf{a}}{\mathbf{a} \cdot \mathbf{a}}\mathbf{a}, \tag{10}$$

where $\mathbf{a} \in \mathbb{R}^n$ is a vector orthogonal to the mirror (normal vector) and $\mathbf{c} \in \mathbb{R}^n$ is a point through which the mirror passes. $\mathbf{a}$ and $\mathbf{c}$ are parameters that determine the mirror.

### 4.3 REFLECTION-BASED WORD ATTRIBUTE TRANSFER

We apply reflection to the word attribute transfer to invert a specific attribute of an input word without its explicit attribute knowledge. We learn a mirror (hyperplane) in a pre-trained embedding space using training word pairs with a common (binary) attribute $\mathbf{z}$ (Fig. 2). Here since the mirror is uniquely determined by two parameter vectors, $\mathbf{a}$ and $\mathbf{c}$, we estimate $\mathbf{a}$ and $\mathbf{c}$ from target attribute $\mathbf{z}$ using fully connected multi-layer perceptrons:

$$\mathbf{a} = MLP(\mathbf{z}), \tag{11}$$

$$\mathbf{c} = MLP(\mathbf{z}). \tag{12}$$

Transferred vector $\mathbf{v}_y$ is obtained by inverting attribute $\mathbf{z}$ of $\mathbf{v}_x$ by reflection:

$$\mathbf{v}_y = Ref_{\mathbf{a},\mathbf{c}}(\mathbf{v}_x). \tag{13}$$

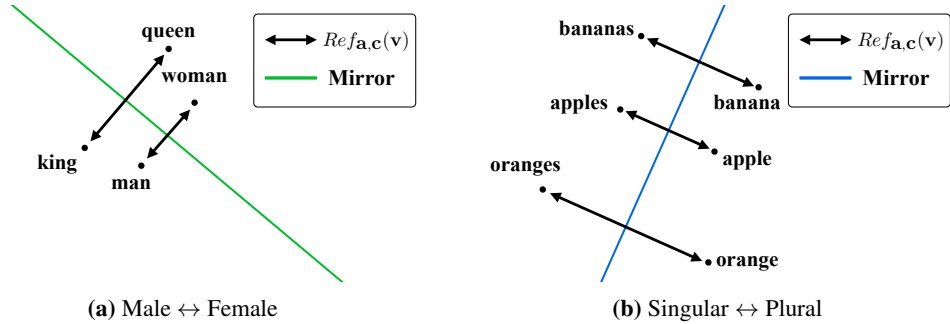

**(a)** Male ↔ Female     **(b)** Singular ↔ Plural

**Figure 2:** Reflection-based word attribute transfer examples.

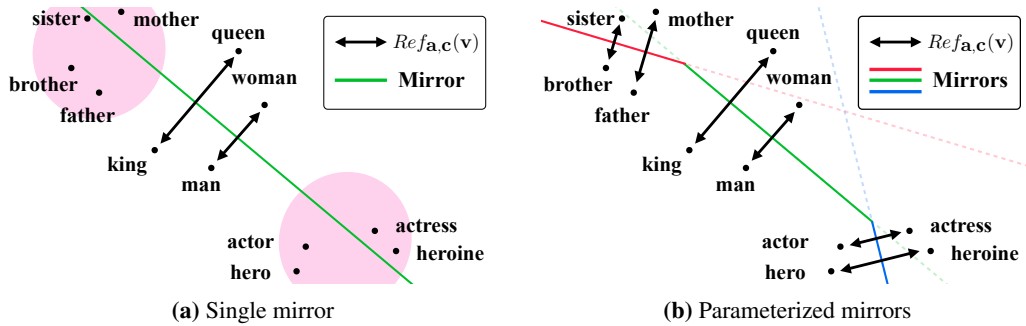

**(a)** Single mirror     **(b)** Parameterized mirrors

**Figure 3:** Mirror estimation methods

## 4.4 PARAMETERIZED MIRRORS

Reflection with a mirror by Eqs. 11 and 12 assumes a single mirror depending only on **z**. Previous discussion assumed that there will be pairs sharing a stable attribute such *king* and *queen*. However, often gendered words don't come in pairs, and gender is far from a stable attribute. For example, *actress* may be feminine, but *actor* is clearly neutral in many cases (Fig. 3). Thus, *actor* isn't as obvious a masculine counterpart as *king*. In fact, it is known that there are biases in gender words in the embedding space (Zhao et al., 2018; Kaneko & Bollegala, 2019). This phenomenon can occur not only with the gender attribute, but also with other attributes. With this assumption of a single mirror, the mirror must be a hyperplane that goes through the midpoints for all word vector pairs. However, the vector pairs shown on the left of Fig. 3 cannot be transferred well since the single mirror does not satisfy this constraint due to the bias of the embedding space. To solve this problem, we introduce different mirrors for different words. We propose *parameterized mirrors* determined by input vector $\mathbf{v}_x$ in addition to attribute **z**. The following are the definitions of the mirror parameters:

$$\mathbf{a} = MLP([\mathbf{z}; \mathbf{v}_x]), \tag{14}$$

$$\mathbf{c} = MLP([\mathbf{z}; \mathbf{v}_x]), \tag{15}$$

where $[\cdot; \cdot]$ indicates the vector concatenation in the column. The *parameterized mirrors* are expected to work flexibly on different words. For instance, as shown in Fig. 3, suppose we learned the mirror (the blue line) that transfers $\mathbf{v}_{hero}$ to $\mathbf{v}_{heroine}$ in advance. If input word vector $\mathbf{v}_{actor}$ resembles $\mathbf{v}_{hero}$, a mirror that is similar to the one for $\mathbf{v}_{hero}$ should be derived and used for the attribute transfer.

## 4.5 LOSS FUNCTION

Loss function $\mathcal{L}$ is defined:

$$\mathcal{L}(\Theta) = \frac{1}{|\mathcal{A}|} \sum_{(x_i, t_i, \mathbf{Z}_i) \in \mathcal{A}} (\mathbf{v}_{y_i} - \mathbf{v}_{t_i})^2 + \frac{1}{|\mathcal{N}|} \sum_{x_j \in \mathcal{N}} (\mathbf{v}_{y_j} - \mathbf{v}_{x_j})^2, \tag{16}$$

where $\frac{1}{|\mathcal{A}|} \sum_{(x_i,t_i,\mathbf{z}_i)\in\mathcal{A}} (\mathbf{v}_{y_i} - \mathbf{v}_{t_i})^2$ is a term that draws target word vector $\mathbf{v}_{t_i}$ closer to corresponding transferred vector $\mathbf{v}_{y_i}$ and $\frac{1}{|\mathcal{N}|} \sum_{x_j\in\mathcal{N}} (\mathbf{v}_{y_j} - \mathbf{v}_{x_j})^2$ is a term that prevents words without a target attribute from being moved by transfer function $f_{\mathbf{Z}}$. $\Theta$ represents the set of all the trainable parameters. The parameters in the proposed model are the MLP weights used to determine mirror hyperplanes via $\mathbf{a}$ and $\mathbf{c}$. We iteratively update $\Theta$ to minimize $\mathcal{L}$:

$$\Theta_{t+1} \leftarrow \arg\min_{\Theta_t} \mathcal{L}(\Theta_t), \tag{17}$$

where $t$ is the number of parameter updates at that time.

## 5 EXPERIMENT

We evaluated the performance of the proposed reflection-based word attribute transfer using data with some different attributes.

### 5.1 EXPERIMENTAL SETUP

We used three different datasets of word pairs with three binary attributes: Male-Female [1], Singular-Plural and Capital-Country, shown in Table 1.These word pairs were collected from analogy test sets (Mikolov et al., 2013a; Gladkova et al., 2016), the Internet and Nguyen et al. (2017) for antonyms of noun[2]. Since these datasets are very small, we added Gaussian noise ($\sigma = 0.1$) to every input vector $\mathbf{v}_x$ during training to avoid overfit. Random noise was applied independently to every sample in every iteration. For a non-attribute dataset $\mathcal{N}$, we sampled words from the three-million-word vocabulary of the word embedding model. We sampled from 4 to 50 words for training ($0 \leq |\mathcal{N}_{\text{train}}| \leq 50$) and 1000 words for the test ($|\mathcal{N}_{\text{test}}| = 1000$). We used a mixed dataset that included both $|\mathcal{N}_{\text{train}}|$ and $|\mathcal{A}_{\text{train}}|$. Note that $\mathcal{N}_{\text{test}}$ was sampled and excluded words from $\mathcal{N}_{\text{train}}$ and $\mathcal{A}_{\text{train}}$. We had no $\mathcal{N}_{\text{val}}$ because the tuning was conducted with only $|\mathcal{A}_{\text{val}}|$. We used word2vec (Mikolov et al., 2013b) [3] and GloVe (Pennington et al., 2014) [4] as the pre-trained embedding model. The embedded vector dimension is $n = 300$.

**Table 1:** Statistics of binary attribute word pair datasets (in the number of word pairs).

| Dataset $\mathcal{A}$ | Train | Val | Test | Total |
|---|---|---|---|---|
| Male-Female (MF) | 29 | 12 | 12 | 53 |
| Singular-Plural (SP) | 90 | 25 | 25 | 140 |
| Capital-Country (CC) | 59 | 25 | 25 | 109 |
| Antonym (AN) | 1354 | 290 | 290 | 1934 |

### 5.2 EVALUATION METRICS

We measured the accuracy and stability performances of the word attribute transfer. The accuracy measures how many input words in $\mathcal{A}_{\text{test}}$ were transferred correctly to the corresponding target words. The stability score measures how many words in $\mathcal{N}_{\text{test}}$ are not mapped to other words. For example, in a gender transfer, given *man*, the transfer is regarded as correct if *woman* is the closest word to the transferred vector; otherwise it is incorrect. Given *apple*, the transfer is regarded as correct if *apple* is the closest word to the transferred vector; otherwise its stability is incorrect. Here we used cosine similarity to measure the similarity of output vector $\mathbf{v}_y$ and target vector $\mathbf{v}_t$. The accuracy and stability scores are calculated by the following formula:

$$\delta(\mathbf{v}_y, t) = \begin{cases} 1 & \text{if} \quad \arg\max_{k\in\mathcal{V}} \frac{\mathbf{v}_y \cdot \mathbf{v}_k}{\|\mathbf{v}_y\|\|\mathbf{v}_k\|} = t, \\ 0 & \text{otherwise,} \end{cases} \tag{18}$$

---

[1]Note that these gender word pairs are an assumption because they contain socially problematic words.
[2]These datasets and the experimental codes will be released in the future.
[3]https://code.google.com/archive/p/word2vec/
[4]https://nlp.stanford.edu/projects/glove/

$$\text{Accuracy} = \frac{1}{|\mathcal{A}_{\text{test}}|} \sum_{(x_i, t_i, \mathbf{Z}_i) \in \mathcal{A}_{\text{test}}} \delta(\mathbf{v}_{y_i}, t_i), \tag{19}$$

$$\text{Stability} = \frac{1}{|\mathcal{N}_{\text{test}}|} \sum_{x_i \in \mathcal{N}_{\text{test}}} \delta(\mathbf{v}_{y_i}, x_i), \tag{20}$$

where $\mathcal{V}$ is the vocabulary of the word embedding model.

## 5.3 METHODS AND CONFIGURATIONS

In our experiment, we compared our proposed method with the following baseline methods:

REF  Reflection-based word attribute transfer with a single mirror. We used a fully connected 2-layer MLP with 300 hidden units and ReLU activations to estimate $\mathbf{a}$ and $\mathbf{c}$.

REF+PM  Reflection-based word attribute transfer with *parameterized mirrors*. We used the same MLP as the REF.

MLP  Fully connected MLP: $\mathbf{v}_y = MLP([\mathbf{v}_x; \mathbf{z}])$. The highest accuracy models are a 2-layer MLP for Capital-Country and 3-layer MLP for the other datasets.

DIFF  Analogy-based word attribute transfer with a difference vector: $\mathbf{d} = \mathbf{v}_m - \mathbf{v}_w$, where $m$ and $w$ are in the training data of $\mathcal{A}$. We chose $\mathbf{d}$ because it achieved the best accuracy in the validation data of $\mathcal{A}$. We determined whether to add or subtract $\mathbf{d}$ to $\mathbf{v}_x$ based on attribute knowledge (Eq. 4).

DIFF $^+$  Analogy-based word attribute transfer with a difference vector regardless of the attribute knowledge. $\mathbf{d}$ was obtained in the same way as the DIFF. We added $\mathbf{d}$ to $\mathbf{v}_x$ for any input $x$: $f_{\mathbf{Z}}(\mathbf{v}_x) = \mathbf{v}_x + \mathbf{d}, \quad \forall \mathbf{v}_x \in \mathbb{R}^n$.

DIFF $^-$  Analogy-based word attribute transfer with a difference vector regardless of the attribute knowledge. $\mathbf{d}$ was obtained in the same way as the DIFF. We subtracted $\mathbf{d}$ from $\mathbf{v}_x$ for any input $x$: $f_{\mathbf{Z}}(\mathbf{v}_x) = \mathbf{v}_x - \mathbf{d}, \quad \forall \mathbf{v}_x \in \mathbb{R}^n$.

MEANDIFF  Analogy-based word attribute transfer with a mean difference vector $\bar{\mathbf{d}}$: $\bar{\mathbf{d}} = \frac{1}{|\mathcal{A}_{\text{train}}|} \sum_{(m_i, w_i) \in \mathcal{A}_{\text{train}}} (\mathbf{v}_{m_i} - \mathbf{v}_{w_i})$. We determined whether to add or subtract $\bar{\mathbf{d}}$ to $\mathbf{v}_x$ based on the attribute knowledge (Eq.4).

MEANDIFF $^+$  Analogy-based word attribute transfer with a mean difference vector regardless of the attribute knowledge: $f_{\mathbf{Z}}(\mathbf{v}_x) = \mathbf{v}_x + \bar{\mathbf{d}}, \quad \forall \mathbf{v}_x \in \mathbb{R}^n$. $\bar{\mathbf{d}}$ was obtained in the same way as the MEANDIFF.

MEANDIFF $^-$  Analogy-based word attribute transfer with a mean difference vector regardless of the attribute knowledge: $f_{\mathbf{Z}}(\mathbf{v}_x) = \mathbf{v}_x - \bar{\mathbf{d}}, \quad \forall \mathbf{v}_x \in \mathbb{R}^n$. $\bar{\mathbf{d}}$ was obtained in the same way as the MEANDIFF.

Based on the tuning, we used the Adam optimizer (Kingma & Ba, 2015) with a learning rate of $\alpha = 10^{-4}$ (the other hyperparameters were the same as the original one (Kingma & Ba, 2015)), and a batchsize of 62 for male-female, and 32 for the others. These hyperparameters were identical for the learning-based methods: REF, REF + PM, or MLP. We did not use such regularization methods as dropout (Srivastava et al., 2014) or batch normalization (Ioffe & Szegedy, 2015) because they did not show any improvement in our pilot test.

## 5.4 ACCURACY AND STABILITY

Table 2 and 3 shows the transfer accuracy and stability score results. Both experiments using GloVe or word2vec obtained similar results. REF + PM achieved the best accuracy among the methods that did not use explicit attribute knowledge. This means that reflection can be used for word attribute transfers even without attribute knowledge. In the stability evaluation, reflection-based methods (REF, REF + PM) and the analogy-based methods with a mean difference vector (MEANDIFF $^-$, MEANDIFF $^+$) achieved high stability. In particular, reflection-based transfers achieved outstanding stability scores exceeding 99%. The stability of DIFF $^+$ and DIFF $^-$ was much lower than the other methods. Although MEANDIFF $^-$ and MEANDIFF $^+$ achieved high stability, their accuracy results

**Table 2:** Results in accuracy and stability scores (word2vec).

| Method | Knowledge | Accuracy (%) | | | | Stability (%) | | | |
|---|---|---|---|---|---|---|---|---|---|
| | | MF | SP | CC | AN | MF | SP | CC | AN |
| REF | | 20.83 | 0.00 | 36.00 | 0.00 | 99.80 | **100.00** | 99.80 | **100.00** |
| REF + PM | | **41.67** | **22.00** | **58.00** | 28.79 | **99.90** | 99.40 | 99.40 | **100.00** |
| MLP | | 8.33 | 4.00 | 12.00 | **35.86** | 2.20 | 0.00 | 2.70 | 1.90 |
| DIFF $^+$ | | 25.00 | 2.00 | 32.00 | - | 72.10 | 77.90 | 53.90 | - |
| DIFF $^-$ | | 25.00 | 2.00 | 30.00 | - | 49.60 | 78.20 | 56.30 | - |
| MEANDIFF $^+$ | | 4.17 | 0.00 | 22.00 | - | 98.60 | 99.40 | 87.60 | - |
| MEANDIFF $^-$ | | 8.33 | 0.00 | 14.00 | - | 97.20 | 99.30 | 92.40 | - |
| DIFF | ✓ | 62.50 | 4.00 | 64.00 | - | - | - | - | - |
| MEANDIFF | ✓ | 12.50 | 0.00 | 36.00 | - | - | - | - | - |

**Table 3:** Results in accuracy and stability scores (GloVe).

| Method | Knowledge | Accuracy (%) | | | | Stability (%) | | | |
|---|---|---|---|---|---|---|---|---|---|
| | | MF | SP | CC | AN | MF | SP | CC | AN |
| REF | | 12.50 | 2.00 | 26.00 | 0.00 | **100.00** | **100.00** | **100.00** | **100.00** |
| REF + PM | | **45.83** | **50.00** | **76.00** | 33.54 | 99.70 | 99.10 | 99.20 | **100.00** |
| MLP | | 4.17 | 10.00 | 18.00 | **36.72** | 5.10 | 7.00 | 5.20 | 1.20 |
| DIFF $^+$ | | 25.00 | 2.00 | 26.00 | - | 99.30 | 94.20 | 99.30 | - |
| DIFF $^-$ | | 25.00 | 2.00 | 24.00 | - | 100.60 | 99.90 | 99.50 | - |
| MEANDIFF $^+$ | | 0.00 | 0.00 | 22.00 | - | **100.00** | **100.00** | **100.00** | - |
| MEANDIFF $^-$ | | 0.00 | 0.00 | 0.00 | - | **100.00** | **100.00** | **100.00** | - |
| DIFF | ✓ | 50.00 | 4.00 | 44.00 | - | - | - | - | - |
| MEANDIFF | ✓ | 0.00 | 0.00 | 0.00 | - | - | - | - | - |

were very low. Interestingly, reflection-based transfer with *parameterized mirrors* (REF + PM) achieved high performance in both accuracy and stability. For example, the accuracy of REF + PM was 41.67%, and the stability was 99.9% in Male-Female (MF), and the accuracy was 58 % and the stability was 99.40 % in Capital-Country (CC). These results show that the proposed method transfers an input word if it has a target attribute and does not transfer an input word even though it does not use explicit attribute knowledge on the input words. MLP worked poorly both in accuracy and stability. In the antonym (AN), while the transfer accuracy by the proposed method was a bit lower than that by MLP, the stability of the proposed method was 100% and that of MLP was really poor (almost 0%).

We investigated the relation between the size of $|\mathcal{N}_{\text{train}}|$ and the stability of learning-based methods by conducting an additional experiment by varying $|\mathcal{N}_{\text{train}}|$ from 0 to 50. The stability scores by MLP did not improve (Table 4). On the other hand, REF and REF + PM achieved high stability scores with just $|\mathcal{N}_{\text{train}}| = 4$ and maintained the accuracy.

## 5.5 TRANSFER EXAMPLE

Table 5 shows examples of a gender transfer at the sentence level, where the attribute transfer was applied to words in sentence $X = \{x_1, x_2, ...\}$. Here since such words as *a* and . are not in the vocabulary of the original word embedding model, we omitted them from the inputs. MLP made many wrong transfers on words without gender attributes, e.g., *the* became *By_Katie_Klingsporn*, *was* became *she*, *when* became *Doughty_Evening_Chronicle*, and *woman* became *girlfriend*. DIFF$^+$ can transfer if $x$ is female, e.g., it transferred from *woman* to *man*, but it could not transfer *grandfather* and *boy*. Similarly, DIFF $^-$ failed to transfer from female to male. In addition, since the stability of these methods was low, they erroneously transferred. For example, in DIFF $^-$, *the* and *when* became *she*. REF + PM can selectively transfer words with a gender attribute without using explicit gender information. For example, when *woman* was given, $Ref(\mathbf{v}_{woman})$ became *man* without knowledge that *woman* is a female word, and when *man* was given, it became *woman*. When non-attribute word

**Table 4:** Relation among size of $|\mathcal{N}_{\text{train}}|$ and stability of learning-based methods.

| | | Accuracy (%) | | | | Stability (%) | | | |
|---|---|---|---|---|---|---|---|---|---|
| | | $|\mathcal{N}_{\text{train}}|$ | | | | $|\mathcal{N}_{\text{train}}|$ | | | |
| | | 0 | 4 | 10 | 50 | 0 | 4 | 10 | 50 |
| **MF** | REF | 16.67 | 20.83 | 20.83 | 20.83 | **98.30** | **98.80** | 99.40 | 99.80 |
| | REF + PM | **41.67** | **45.83** | 20.83 | **41.67** | 38.30 | 98.40 | **100.00** | **99.90** |
| | MLP | 4.17 | 8.33 | 8.33 | 8.33 | 0.00 | 0.30 | 0.30 | 2.20 |
| **SP** | REF | 0.00 | 0.00 | 0.00 | 0.00 | **99.90** | **99.90** | **99.90** | 100.90 |
| | REF + PM | **12.00** | **22.00** | **18.00** | **18.00** | 98.40 | 99.40 | 99.30 | 99.80 |
| | MLP | 4.00 | 4.00 | 2.00 | 2.00 | 0.00 | 0.00 | 0.10 | 3.40 |
| **CC** | REF | 36.00 | 36.00 | 36.00 | 34.00 | **99.80** | **99.80** | **99.80** | 100.00 |
| | REF + PM | **58.00** | **56.00** | **58.00** | **54.00** | 73.80 | 99.70 | 99.40 | 99.40 |
| | MLP | 6.00 | 6.00 | 8.00 | 12.00 | 0.00 | 0.30 | 0.50 | 2.70 |
| **AN** | REF | 0.00 | 0.00 | 0.00 | 0.00 | **99.90** | **99.90** | **99.90** | 99.80 |
| | REF + PM | 21.72 | 27.24 | 28.62 | 28.79 | 95.30 | 99.20 | 99.50 | **99.80** |
| | MLP | **34.14** | **35.00** | **34.31** | **35.86** | 0.00 | 0.01 | 0.02 | 1.90 |

*married* was given, $Ref(\mathbf{v}_{married})$ became *married* without knowledge that *married* has no gender attribute. When we applied the reflection-based transfer twice, the transferred word returned to its original word, e.g., $Ref(Ref(\mathbf{v}_{woman}))$ gives *woman*.

**Table 5:** Transfer results when sentence $X = \{\text{the}, ..., \text{boy}\}$ was given. Out-of-vocabulary words *a* and *.* were not given as input.

| $X$ | the woman was married when your grandfather was (a) boy (.) |
|---|---|
| $Ref(x)$ | the **man** was married when your **grandmother** was (a) **girl** (.) |
| $Ref(Ref(x))$ | the **woman** was married when your **grandfather** was (a) **boy** (.) |
| MLP | By_Katie_Klingsporn **girlfriend** she fiancee Doughty_Evening_Chronicle ma'am **daughter** she (a) **mother** (.) |
| DIFF $^+$ | the **man** was married when your **grandfather** was (a) **boy** (.) |
| DIFF $^-$ | she **woman** was married she your **grandmother** was (a) **girl** (.) |

We can transfer some different attributes of words with reflection-based transfer one-by-one. Table 6 shows that the words having different target attributes were transferred by each reflection-based transfer in the order of Male-Female, Singular-Plural, and Country-Capital. Given *actress* for a Male-Female transfer, it was transferred to *actor* and to *actors* for Singular-Plural. Given *Tokyo* for Male-Female, Singular-Plural, and Antonym, it was not transferred, but it was transferred to *Japan* for Country-Capital. Given *rich* for Male-Female, Singular-Plural, and Capital-Country, it was not transferred, but it was transferred to *poor* for Antonym.

**Table 6:** Transfer of different attributes with reflection-based word attribute transfer with *parameterized mirrors*.

| $X$ | the rich actress and the poor actor want to stay the beautiful city in Tokyo. |
|---|---|
| + Male-Female | the rich **actor** and the poor **actress** want to stay the beautiful city in Tokyo. |
| + Singular-Plural | the rich **actors** and the poor **actresses** want to stay the beautiful **cities** in Tokyo. |
| + Capital-Country | the rich actors and the poor actresses want to stay the beautiful citie in **Japan**. |
| + Antonym | the **poor** actors and the **rich** actresses want to stay the beautiful cities in Japan. |

## 6 RELATED WORK

The embedded vectors obtained by SGNS (Mikolov et al., 2013a;b) and GloVe (Pennington et al., 2014) have analogic relations. The theory of analogic relations in word embeddings has been widely discussed: Levy & Goldberg (2014b); Arora et al. (2016); Gittens et al. (2017); Ethayarajh et al. (2019); Allen & Hospedales (2019); Linzen (2016). Levy & Goldberg (2014b) offer the explanation that SGNS factorizes a shifted PMI matrix. Allen & Hospedales (2019) and Ethayarajh et al. (2019) argued that they proved the existence of such analogic relations without strong assumptions. In our work, we focus on the analogic relations in a word embedding space and propose a novel framework to obtain a word vector with inverted attributes. Style transfers (Niu et al., 2018; Prabhumoye et al., 2018; Jain et al., 2019; Logeswaran et al., 2018; Dai et al., 2019; Zhang et al., 2018) resemble our task. In a style transfer, the text style of the input sentences is changed. For instance, Jain et al. (2019) transferred from formal to informal sentences. Logeswaran et al. (2018) transferred sentences by controlling such attributes as mood and tense. These style transfer tasks use sentence pairs; our word attribute transfer task uses word pairs. Style transfer changes sentence styles, but our task changes the word attributes (contents). Soricut & Och (2015) studied the problem of morphological transformation based on character information. Our work aims more general attribute transfer such as gender transfer and country-capital and is not limited to the morphological transformation.

## 7 CONCLUSION AND FUTURE WORK

We proposed a novel representation learning framework based on reflection to invert a certain attribute of a word vector. We proposed a reflection-based method for word attribute transfers without relying on the explicit attribute knowledge of an input word, which is necessary for a simple analogy-based transfer. Experimental results showed that our proposed method can transfer the word attributes if the input word has a target attribute. If not, reflection does not transfer the word.

Future work includes applications to other transfer tasks: sentence by sentence transfer, such Niu et al. (2018); Prabhumoye et al. (2018); Jain et al. (2019), and entity prediction on an analogic graph embedding space (Liu et al., 2017), in the field of computer vision, visual analogy (Reed et al., 2015), or style transfer (Zhu et al., 2017; Liao et al., 2017) with GANs (Radford et al., 2016; Goodfellow et al., 2014) because their latent space holds analogic relations (Radford et al., 2016).

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

# A  TOP THREE ACCURACY AND STABILITY

**Table 7:** Male-Female

| Method | Knowledge | Accuracy (%) | | | | Stability (%) | | | |
|---|---|---|---|---|---|---|---|---|---|
| | | Mean@3 | @1 | @2 | @3 | Mean@3 | @1 | @2 | @3 |
| REF | | 50.00 | 20.83 | **62.50** | **66.67** | **99.80** | **99.80** | **99.80** | **99.80** |
| REF + PM | | **55.56** | **41.67** | 58.33 | **66.67** | 99.13 | 99.00 | 99.20 | 99.20 |
| MLP | | 20.83 | 8.33 | 20.83 | 33.33 | 2.20 | 2.20 | 2.20 | 2.20 |
| DIFF $^+$ | | 31.94 | 25.00 | 33.33 | 37.50 | 75.43 | 72.10 | 75.80 | 78.40 |
| DIFF $^-$ | | 31.94 | 25.00 | 33.33 | 37.50 | 55.80 | 49.60 | 57.30 | 60.50 |
| MEANDIFF $^+$ | | 23.61 | 4.17 | 33.33 | 33.33 | 98.93 | 98.60 | 99.10 | 99.10 |
| MEANDIFF $^-$ | | 23.61 | 8.33 | 29.17 | 33.33 | 97.63 | 97.20 | 97.80 | 97.90 |
| DIFF | ✓ | 68.05 | 62.50 | 66.66 | 75.00 | - | - | - | - |
| MEANDIFF | ✓ | 47.22 | 12.50 | 62.50 | 66.67 | - | - | - | - |

**Table 8:** Singular-Plural

| Method | Knowledge | Accuracy (%) | | | | Stability (%) | | | |
|---|---|---|---|---|---|---|---|---|---|
| | | Mean@3 | @1 | @2 | @3 | Mean@3 | @1 | @2 | @3 |
| REF | | 50.00 | 0.00 | **72.00** | **78.00** | **100.00** | **100.00** | **100.00** | **100.00** |
| REF + PM | | **56.67** | **22.00** | **72.00** | 76.00 | 99.70 | 99.40 | 99.80 | 99.90 |
| MLP | | 7.33 | 4.00 | 8.00 | 10.00 | 0.13 | 0.00 | 0.20 | 0.20 |
| DIFF $^+$ | | 36.67 | 2.00 | 50.00 | 58.00 | 80.27 | 77.90 | 79.70 | 83.20 |
| DIFF $^-$ | | 36.67 | 2.00 | 52.00 | 56.00 | 80.27 | 78.20 | 80.70 | 81.90 |
| MEANDIFF $^+$ | | 42.00 | 0.00 | 56.00 | 70.00 | 99.47 | 99.40 | 99.40 | 99.60 |
| MEANDIFF $^-$ | | 39.33 | 0.00 | 56.00 | 62.00 | 99.50 | 99.30 | 99.60 | 99.60 |
| DIFF | ✓ | 49.33 | 4.00 | 70.00 | 74.00 | - | - | - | - |
| MEANDIFF | ✓ | 52.00 | 0.00 | 76.00 | 80.00 | - | - | - | - |

**Table 9:** Capital-Country

| Method | Knowledge | Accuracy (%) | | | | Stability (%) | | | |
|---|---|---|---|---|---|---|---|---|---|
| | | Mean@3 | @1 | @2 | @3 | Mean@3 | @1 | @2 | @3 |
| REF | | 66.67 | 36.00 | **78.00** | **86.00** | **99.80** | **99.80** | **99.80** | **99.80** |
| REF + PM | | **72.67** | **58.00** | 74.00 | **86.00** | 99.53 | 99.40 | 99.60 | 99.60 |
| MLP | | 35.33 | 12.00 | 40.00 | 54.00 | 2.80 | 2.70 | 2.80 | 2.90 |
| DIFF $^+$ | | 39.33 | 32.00 | 42.00 | 44.00 | 64.87 | 53.90 | 69.60 | 71.10 |
| DIFF $^-$ | | 34.67 | 30.00 | 36.00 | 38.00 | 58.63 | 56.30 | 58.90 | 60.70 |
| MEANDIFF $^+$ | | 36.00 | 22.00 | 42.00 | 44.00 | 89.33 | 87.60 | 89.70 | 90.70 |
| MEANDIFF $^-$ | | 34.67 | 14.00 | 44.00 | 46.00 | 93.07 | 92.40 | 93.10 | 93.70 |
| DIFF | ✓ | 80.00 | 64.00 | 86.00 | 90.00 | - | - | - | - |
| MEANDIFF | ✓ | 70.67 | 36.00 | 86.00 | 90.00 | - | - | - | - |

**Table 10:** Antonym

| Method | Knowledge | Accuracy (%) | | | | Stability (%) | | | |
|---|---|---|---|---|---|---|---|---|---|
| | | Mean@3 | @1 | @2 | @3 | Mean@3 | @1 | @2 | @3 |
| REF | | 0.94 | 0.00 | 1.19 | 16.21 | **99.97** | **99.90** | **100.00** | **100.00** |
| REF + PM | | 38.56 | 28.79 | 41.38 | 45.52 | 99.93 | 99.80 | **100.00** | **100.00** |
| MLP | | **41.38** | **35.86** | **42.59** | **45.69** | 1..97 | 1.90 | 2.00 | 2.00 |

## B  VISUALIZATION OF PARAMETERIZED MIRRORS

Figures below visualize PCA results of $\mathbf{a}$ obtained for the test words. We normalized the L2 norm of $\mathbf{a}$ to 1 ($\frac{\mathbf{a}}{\|\mathbf{a}\|}$). Corresponding word pairs are connected by solid lines. Figs. 4 and 5 suggest not only the mirror parameters of paired words are similar to each other but also the parameters with the attribute form a cluster — words with the same attribute has similar mirror parameter $\mathbf{a}$. The mirrors of paired words are close to each other in the same attribute (Fig. 4 and 5). Some MF pairs in Fig. 5 are placed away from the cluster of the MF words. This may come from missing principal components due to the small data size used for PCA. Figs. 6, 7, 8, 9, 10, 11, 12, and 13 are the detailed PCA results for four different attributes: MF, SP, CC, and AN. These results show that a reflection transfers a paired word each other by using a similar mirror. For example, *rich* and *poor* use almost the same mirror (Fig. 9). On the other hand, different mirrors are used for different word pairs since the mirrors are parameterized. These results shows the effect of the mirror as described in section 4.4.

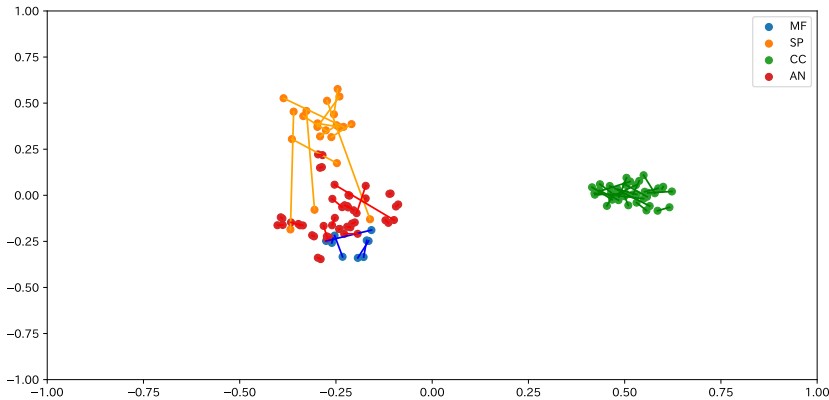

**Figure 4:** A PCA result of $\mathbf{a}$ (word2vec).

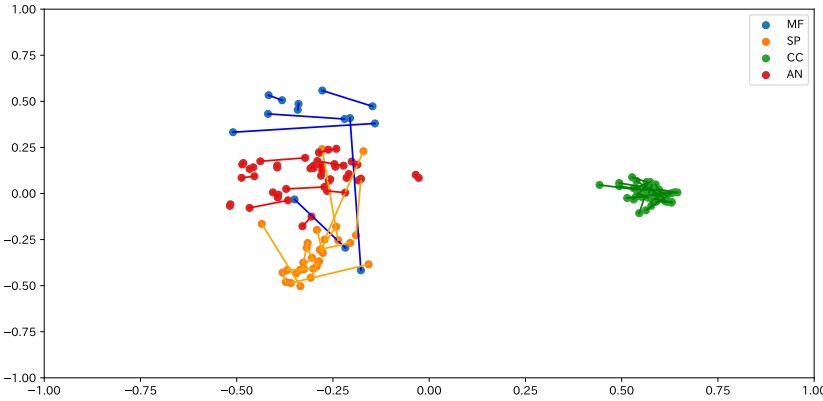

**Figure 5:** A PCA result of $\mathbf{a}$ (GloVe).

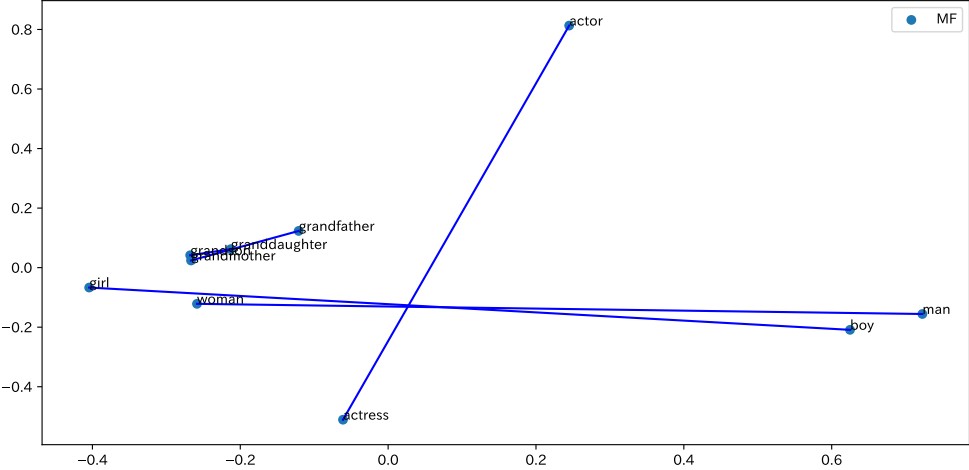

**Figure 6:** Male-Female (word2vec)

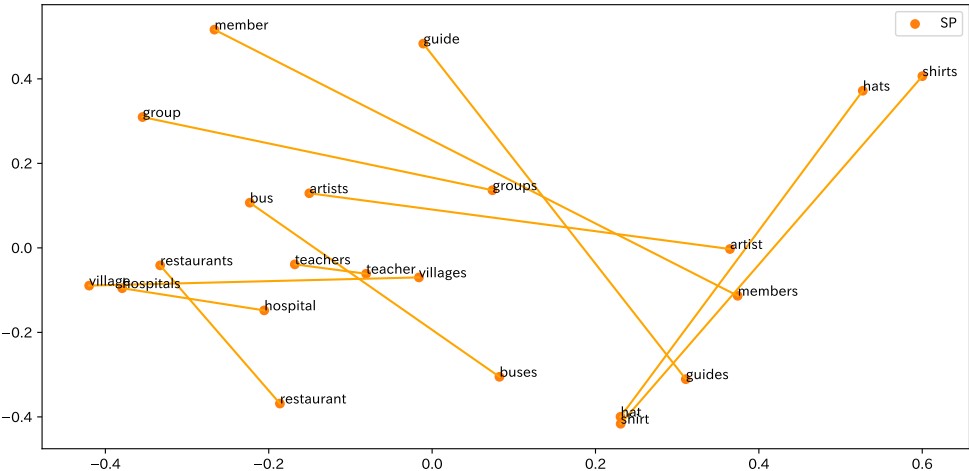

**Figure 7:** Singular-Plural (word2vec)

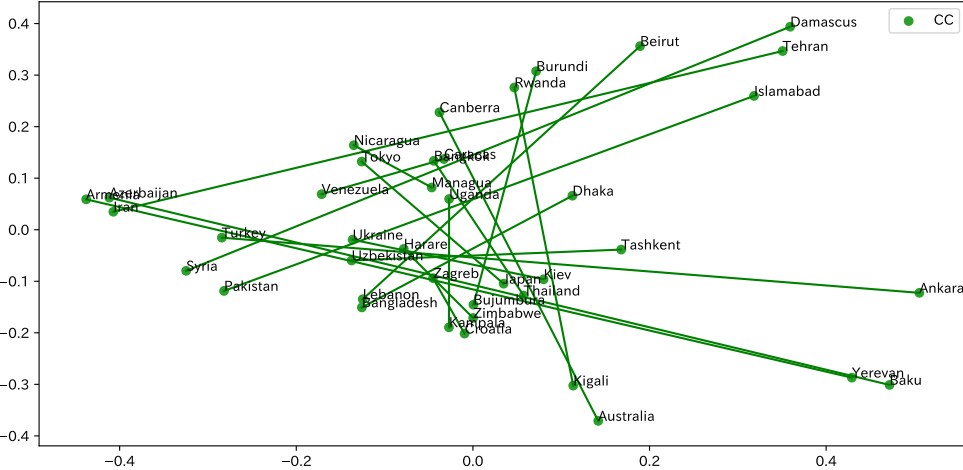

**Figure 8:** Capital-Country (word2vec)

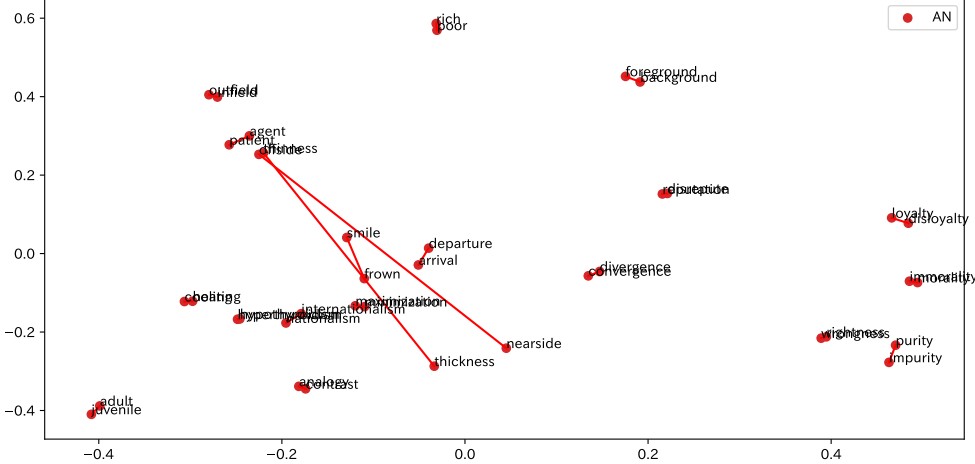

**Figure 9:** Antonym (word2vec)

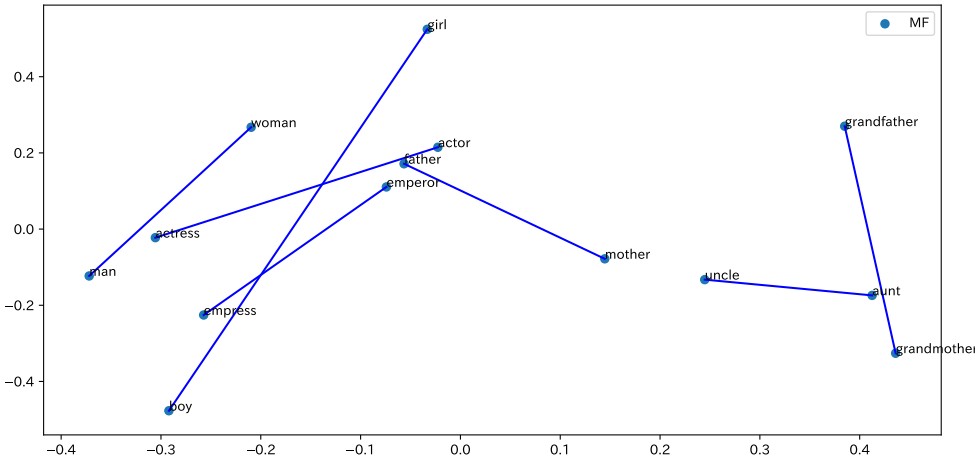

**Figure 10:** Male-Female (GloVe)

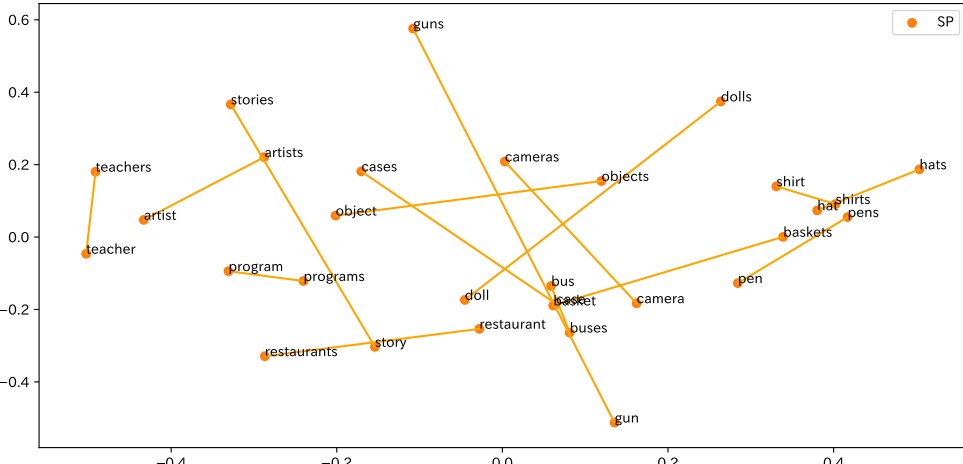

**Figure 11:** Singular-Plural (GloVe)

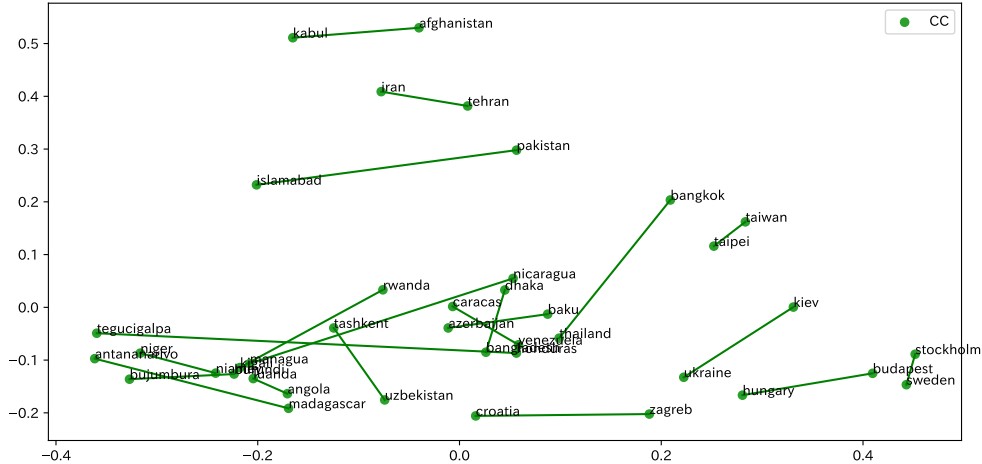

**Figure 12:** Capital-Country (GloVe)

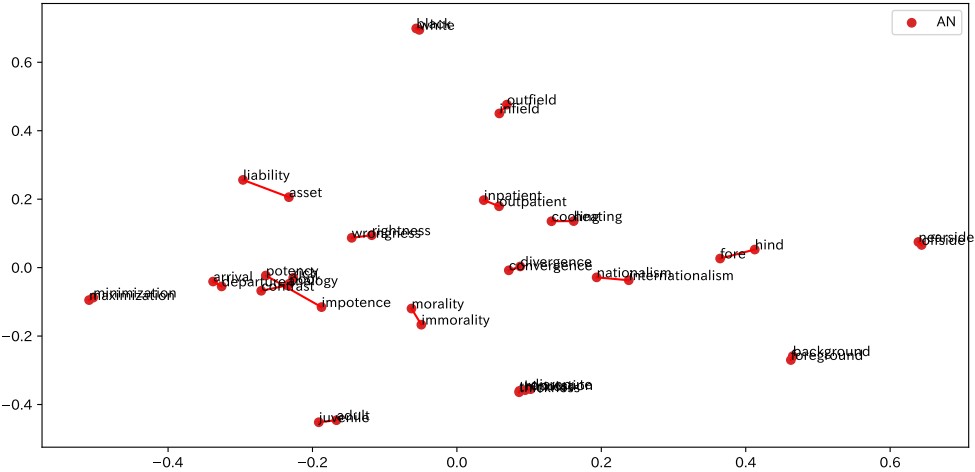

**Figure 13:** Antonym (GloVe)

