# OpenReview forum: "Reflection-based Word Attribute Transfer"
_ICLR.cc/2020/Conference — Reject_

### Official Review · AnonReviewer1 · 2019-10-23
**Official Blind Review #1**

**Rating:** 6

**Review:**

If I understood correctly, in experimental part pre-trained embeddings, taken from word2vec, is a ground truth. Given those embeddings, a system of hyperplanes are trained (every hyperplane refers to a certain word attribute and a region in space, centered around a word, to which reflection is applied).

In my opinion, the most natural way to check "the reflection hypothesis" is to train new embeddings where to word2vec objective the loss function (16) is added and to look how perplexity will behave with that additional cost and to look how your accuracy and stability will behave on the test set.

It is also interesting to learn how this reflection based attribute transfer can be applied to the same word, but with embeddings that play different role in a model: e.g. input and output embeddings.
In fact, input and output embeddings are located in the same space, they can be considered as pairs of words with 1 attribute flipped (i.e. role in a model, input->output). Can an output embedding be obtained from an input embedding via reflections that you describe. How many hyperplanes we need in that case? This is an interesting edge of the problem. In simplest models, there is some evidence, that output embeddings are result of reflections of input embeddings in half the dimensions. It would be interesting to learn your comment on that.


**Experience Assessment:**

I have published one or two papers in this area.

**Review Assessment: Checking Correctness Of Derivations And Theory:**

I assessed the sensibility of the derivations and theory.

**Review Assessment: Checking Correctness Of Experiments:**

I assessed the sensibility of the experiments.

**Review Assessment: Thoroughness In Paper Reading:**

I made a quick assessment of this paper.

---

> ### Author Response · Authors · 2019-11-13
> **Response for Reviewer#1**
>
> Thank you for the encouraging feedback. We address your comments and questions below.
>
> 1. "In my opinion, the most natural way to check “the reflection hypothesis” is to train new embeddings where to word2vec objective the loss function (16) is added and to look how perplexity will behave with that additional cost and to look how your accuracy and stability will behave on the test set."
> In this study, we tried to incorporate reflection into a given word embeddings space as our first step. Training word embeddings from scratch with reflection constraints would be promising future work.
>
> 2. "Can an output embedding be obtained from an input embedding via reflections that you describe. How many hyperplanes we need in that case?"
> Yes, that’s right. In this work, we parameterize mirror hyperplanes instead of a single mirror, and the mirror parameters are determined according to an input word embedding. In other words, we can use different mirror hyperplanes for different input words.

---

### Official Review · AnonReviewer3 · 2019-10-23
**Official Blind Review #3**

**Rating:** 6

**Review:**

Summary: This paper proposes a method for binary word attribute transfer based on reflection. It applies a single reflection-based mapping that relates the locations of two vectors in a Euclidean space by a hyperplane, and results in an identity mapping when it is applied twice. It's sort of like roundtrip-translation, but for word vector attributes. The paper also proposes a pretty smart idea: the mirror functions are parameterized to take advantage of the fact that inversions differ, even for the "same" word attributes.

I am giving the paper a weak accept, because I think idea is really fun, and definitely has legs. I like the idea for this paper quite a bit, it really got me thinking, but I think it isn't quite structured how I would like. Unfortunately, as it is, it also feels a bit spare in terms of contributions. I would like much more discussion/analysis of the parameterizations (and their parameters), or treatment of more binary-attribute linguistic analogy types (antonymy, scalar terms), or both! Another thing that could be added is comparisons across types of word vectors (word2vec alone is a bit narrow). Any of those things would have justified a strong accept.

Suggestions:
- As I was reading, I was not at all convinced by the "idealness" of your transfer function in 4.1 (maybe change it from "ideal" to "idealized"). "Ideal" assumes that there will be pairs sharing a stable attribute. But, often "gendered" words don't come in pairs, and "gender" is far from a stable attribute. To be more specific, despite your Fig 3a: "actress" may be feminine, but "actor" is clearly neutral (anyone can be described as an "actor" but only women can be described as "actress" in most cases); thus "actress" is the gendered one. We can go ahead and pair "actress" with "actor"---they do share a morphological relationship after all---but "actor" isn't as obvious a masculine counterpart as "king" is for "queen". How would you square this with phi(queen)=king and phi(king)=queen, where the identity of counterparts is clearer? Do we ideally want the same invertible method here for both actor-actress and queen-king (I think I don't, hence the function is not "ideal"). I wouldn't be surprised if the same situation arises in other analogies. This is, I think, why you brought up parameterization. But you miss the opportunity to clearly motivate why we need to parameterize! You should include more motivation and lay the linguistic facts out in a clearer way that incorporates examples like the one I gave, and preferably before S4.1.
-The total number of words you chose to use is pretty small, which you acknowledge. I would've loved to see a few more phenomena, perhaps antonymy, part-whole, morphological reinflection (e.g., a generalization of your sing-pl set, datasets exist for past-tense at least), etc.
-it would have been neat to compare word2vec to other types of vector embeddings, maybe contextualized? They are what everyone's using right now (this suggestion didn't factor heavily into my rating though). Also the framing in your introduction made me think this is where you were going (you started with discussing vectors, as opposed to just jumping in on binary word attribute transfer. I think the paper could just jump right in with binary word attribute transfer, and skip the basic vectors paragraph).
-I would've liked more discussion of your parameterized mirrors, since that was the neatest/prettiest part, in my opinion. (maybe I missed something, but how did you decide on how many z to use? or did every pair get its own? Do your zs for each attribute correlate at all? you could use something straightforward like CCA to check, would be neat.)

Small Notes/Musings:
-on p.1 "(PMI (citation..." with nested parentheses is not easy to read, I prefer "(PMI; citation)".
- It is fine to operate on the assumption that gender is binary for now, but you should acknowledge that binary gender is an assumption (and a socially problematic one), that isn't real.
-in 5.5, I'm not impressed by "When non-attribute word the was given, Ref(v the) became 'the' without knowledge that the has no gender attribute", because "the" is the most frequent word in English, perhaps show that it works with rare, non-gendered words instead. Incidentally, does it know syntactic category? Because that might be a neat thing to check.
-what do you think about trinary attributes (cold, warm, hot)? It would be a cool extension.


**Experience Assessment:**

I have read many papers in this area.

**Review Assessment: Checking Correctness Of Derivations And Theory:**

I assessed the sensibility of the derivations and theory.

**Review Assessment: Checking Correctness Of Experiments:**

I assessed the sensibility of the experiments.

**Review Assessment: Thoroughness In Paper Reading:**

I read the paper at least twice and used my best judgement in assessing the paper.

---

> ### Author Response · Authors · 2019-11-13
> **Response for Reviewer#3**
>
> We thank the reviewer for the insightful review and the comments! We address your comments and questions below.
>
> 1. “I would like much more discussion/analysis of the parameterizations (and their parameters), or treatment of more binary-attribute linguistic analogy types (antonymy, scalar terms), or both! Another thing that could be added is comparisons across types of word vectors (word2vec alone is a bit narrow). Any of those things would have justified a strong accept.”
> Thank you for your suggestions. We conducted additional experiments (other embedding, other attributes and an analysis of mirror).  See below responses for details.
>
> 2. “As I was reading, I was not at all convinced by the “idealness” of your transfer function in 4.1 (maybe change it from “ideal” to “idealized”). “
> We changed “ideal” to ““idealized”.
>
> 3. “This is, I think, why you brought up parameterization. But you miss the opportunity to clearly motivate why we need to parameterize! You should include more motivation and lay the linguistic facts out in a clearer way that incorporates examples like the one I gave, and preferably before S4.1.“
> We added such motivation and discussion in Section 4.1.
>
> 4. “The total number of words you chose to use is pretty small, which you acknowledge.  I would’ve loved to see a few more phenomena, perhaps antonymy, part-whole, morphological reinflection (e.g., a generalization of your sing-pl set, datasets exist for past-tense at least), etc.“
> We conducted additional experiments  using antonyms (Table 1) from the study by Nguyen et al.: Distinguishing Antonyms and Synonyms in a Pattern-based Neural Network (EACL2017). While the transfer accuracy by the proposed method was a bit lower than that by MLP, the stability of the proposed method was 100% and that of MLP was really poor (almost 0%).
>
> 5. “it would have been neat to compare word2vec to other types of vector embeddings, maybe contextualized? They are what everyone’s using right now (this suggestion didn’t factor heavily into my rating though). Also the framing in your introduction made me think this is where you were going (you started with discussing vectors, as opposed to just jumping in on binary word attribute transfer. I think the paper could just jump right in with binary word attribute transfer, and skip the basic vectors paragraph).“
> We conducted additional experiments using GloVe and obtained similar results (Table 3). The use of contextualized word embeddings is more difficult in data preparation and would be investigated as our future work.
>
> 6. “I would’ve liked more discussion of your parameterized mirrors, since that was the neatest/prettiest part, in my opinion. (maybe I missed something, but how did you decide on how many z to use? or did every pair get its own? Do your zs for each attribute correlate at all? you could use something straightforward like CCA to check, would be neat.)“
> We used the same $z$ for a word set with the same attribute (e.g., male-female). $z$ for different attributes are independent because one-hot vector $z$ is not trainable vector (e.g., $z =$ [1,0,0]).
>
> 7. “on p.1 '(PMI (citation...' with nested parentheses is not easy to read, I prefer (PMI; citation).“
> Corrected.
>
> 8. “It is fine to operate on the assumption that gender is binary for now, but you should acknowledge that binary gender is an assumption (and a socially problematic one), that isn’t real.“
> We put the acknowledge about our assumption in Section 5.1
>
> 9. “in 5.5, I’m not impressed by “When non-attribute word the was given, Ref(v the) became ‘the’ without knowledge that the has no gender attribute”, because “the” is the most frequent word in English, perhaps show that it works with rare, non-gendered words instead. Incidentally, does it know syntactic category? Because that might be a neat thing to check.“
> The word “the” is just an example. We replaced this example with “married”. Our study does not use any explicit knowledge other than word2vec word vectors, although some syntactic information may be included implicitly.
>
> 10. “what do you think about trinary attributes (cold, warm, hot)? It would be a cool extension.“
> That is a more general and very interesting problem. It requires different kind of transformation and would be investigated in future work.

---

> > ### Author Response · Authors · 2019-11-15
> > **Visualization of parameterized mirrors**
> >
> > Dear reviewer#3,
> >
> > > 6. “I would’ve liked more discussion of your parameterized mirrors, since that was the neatest/prettiest part, in my opinion. (maybe I missed something, but how did you decide on how many z to use? or did every pair get its own? Do your zs for each attribute correlate at all? you could use something straightforward like CCA to check, would be neat.)“
> > > We used the same  for a word set with the same attribute (e.g., male-female).  $z$ for different attributes are independent because one-hot vector $z$ is not trainable vector (e.g.,  [1,0,0]).
> >
> > We visualized the mirror parameter $a$  obtained for the test words instead of $z$, and we have made the revision of the paper (in appendix B).
> > These results suggest not only the mirror parameters of paired words are similar to each other but also the parameters with the attribute form a cluster — words with the same attribute has similar mirror parameter $a$.
> >
> > Best,
> >
> > Authors

---

### Official Review · AnonReviewer2 · 2019-10-23
**Official Blind Review #2**

**Rating:** 6

**Review:**

This paper considers the problem of transferring word attributes between words.  There is a lot of discussion of
elementary ideas such as reflection about an affine plain and involutions.  The key technical idea in the paper is given in the definition of the objective function in equation (16).  The notation in this equation is unfortunate as the parameter vector Theta does not appear in the loss expression and some guesswork is needed to realize that y_i is hat{t}_\Theta(x_i).  The discussion of optimization seems strange --- why not just apply a standard optimizer to this loss function?  The experimental results are limited to three attributes one of which --- the capital/country attribute --- seems a relation not an attribute.

But the most serious problem with this paper is a lack of references to related work.  I would start with the following reference and track papers that reference it.

"Unsupervised Morphology Induction Using Word Embeddings" by Radu Soricut, Franz Och, NAACL 2015.

Postscript: I have read the rebuttal and looked at the revised paper. The citation I suggested has been added but with inadequate acknowledgement.  Two of four attributes studied in the revised paper are morphological (gender is morphological in many languages) and the country-capital relation and antonym relation do not seem to be attributes to me.

On other hand, the results do look promising.  Based entirely on the results I have raised my score weak accept.

**Experience Assessment:**

I have read many papers in this area.

**Review Assessment: Checking Correctness Of Derivations And Theory:**

I assessed the sensibility of the derivations and theory.

**Review Assessment: Checking Correctness Of Experiments:**

I assessed the sensibility of the experiments.

**Review Assessment: Thoroughness In Paper Reading:**

I made a quick assessment of this paper.

---

> ### Author Response · Authors · 2019-11-13
> **Response for Reviewer#2**
>
> Thank you for the valuable review. We address your comments and questions below.
>
> 1. “The notation in this equation is unfortunate as the parameter vector Theta does not appear in the loss expression and some guesswork is needed to realize that y_i is hat{t}_\Theta(x_i).”
> $\Theta$ represents the set of all the trainable parameters. The parameters in the proposed model are the MLP weights used to determine mirror hyperplanes via $a$ and $c$. We added the definition of $\Theta$ in the paper to make it clear.
>
> 2. “The discussion of optimization seems strange --- why not just apply a standard optimizer to this loss function?”
> What do you mean by a standard optimizer? We used Adam in this work as described in the latter part of Section 5.3.
>
>
> 3. “But the most serious problem with this paper is a lack of references to related work.  I would start with the following reference and track papers that reference it.”
> Thank you for the important suggestion. We now mention the work you mentioned in the paper. That work studied the problem of morphological transformation based on character information. Our work aims more general attribute transfer such as gender transfer and country-capital and is not limited to the morphological transformation.

---

### Decision · Program_Chairs · 2019-12-19

**Decision:**

Reject

**Comment:**

This paper proposes a way to transform word vectors based on a binary attribute (e.g. male/female) based on reflection, with the property that applying the reflection operator twice, the vector for a word is left unchanged.  By identifying parameterized mirror planes for each word, the proposed method can leave neutral words left unchanged.

The paper received 3 weak accepts.  There was initially one reject, but the revisions convinced the reviewer to update their score to a weak accept.  Overall, the reviewers appreciated the idea of reflection-based binary word attribute transfer.   suggestions, the authors made small improvements to the writing, added missing citations, as well as additional results for another word embedding (GloVE) and another dataset (antonyms).  One of the main remaining weakness of the work, is still the small dataset.  Although somewhat alleviated by the inclusion of the antonym dataset, this is still a weakness of the paper.

The AC agrees that the paper has an nice idea and is well presented.  However, the work is limited in scope and is likely to be of limited interest to the ICLR community and would be more appreciated in the NLP community.  The authors are encouraged to improve upon the work, and resubmit to an appropriate venue.